# Revisiting the Roles of Filaggrin in Atopic Dermatitis

**DOI:** 10.3390/ijms23105318

**Published:** 2022-05-10

**Authors:** Verena Moosbrugger-Martinz, Corinne Leprince, Marie-Claire Méchin, Michel Simon, Stefan Blunder, Robert Gruber, Sandrine Dubrac

**Affiliations:** 1Department of Dermatology, Venereology and Allergology, Medical University of Innsbruck, Anichstraße 35, 6020 Innsbruck, Austria; verena.martinz@i-med.ac.at (V.M.-M.); stefan.blunder@i-med.ac.at (S.B.); robert.gruber@tirol-kliniken.at (R.G.); 2Toulouse Institute for Infectious and Inflammatory Diseases (Infinity), Toulouse University, CNRS UMR5051, Inserm UMR1291, UPS, 31059 Toulouse, France; corinne.leprince@inserm.fr (C.L.); marie-claire.mechin@inserm.fr (M.-C.M.); michel.simon@inserm.fr (M.S.)

**Keywords:** atopic dermatitis, filaggrin, epidermis, NMF, pH, microbiome

## Abstract

The discovery in 2006 that loss-of-function mutations in the filaggrin gene (*FLG*) cause ichthyosis vulgaris and can predispose to atopic dermatitis (AD) galvanized the dermatology research community and shed new light on a skin protein that was first identified in 1981. However, although outstanding work has uncovered several key functions of filaggrin in epidermal homeostasis, a comprehensive understanding of how filaggrin deficiency contributes to AD is still incomplete, including details of the upstream factors that lead to the reduced amounts of filaggrin, regardless of genotype. In this review, we re-evaluate data focusing on the roles of filaggrin in the epidermis, as well as in AD. Filaggrin is important for alignment of keratin intermediate filaments, control of keratinocyte shape, and maintenance of epidermal texture via production of water-retaining molecules. Moreover, filaggrin deficiency leads to cellular abnormalities in keratinocytes and induces subtle epidermal barrier impairment that is sufficient enough to facilitate the ingress of certain exogenous molecules into the epidermis. However, although *FLG* null mutations regulate skin moisture in non-lesional AD skin, filaggrin deficiency per se does not lead to the neutralization of skin surface pH or to excessive transepidermal water loss in atopic skin. Separating facts from chaff regarding the functions of filaggrin in the epidermis is necessary for the design efficacious therapies to treat dry and atopic skin.

## 1. Filaggrin: A Look in the Rear-View Mirror

Filaggrin was initially isolated from a protein fraction of the stratum corneum (SC) and identified as a basic histidine-rich protein [1]. Proteins from this fraction were shown to aggregate with keratin filaments to form macrofibrils in in vitro cell-free experiments [2]. In 1981, Dale et al. designated this class of cationic structural proteins, which associate specifically with intermediate filaments but not with other types of cytoskeletal proteins, as filaggrins (for filament aggregating proteins) [2]. They showed that filaggrins are species-distinct proteins; for example, rat and mouse filaggrins have different molecular weights (48 and 30 kDa, respectively) and different amino acid totals but nevertheless exhibit similar chemical and functional properties. The ability of filaggrin to aggregate keratin filaments into tight parallel arrays [3] has been demonstrated in many different experimental settings, with high reproducibility [4,5,6,7]. This molecular bundling confers mechanical resilience and flexibility to the SC [8]. Mouse and human filaggrin bind to each three-chain building block of the intermediate filaments, possibly through ionic interactions with the coiled-coil alpha-helical regions of the keratin filaments. The stoichiometry of the interaction has been reported as two filaggrin molecules to three intermediate (keratin) filament subunits [3,9].

Further work demonstrated that filaggrin is produced as a phosphorylated precursor, profilaggrin, of 300–500 kDa, depending on the species [10,11,12], embedded in keratohyalin granules in granular keratinocytes (KCs) [2,13,14]. Subsequently, profilaggrin undergoes a multiple-step dephosphorylation process, followed by proteolytic cleavage of short linker peptides to produce various numbers of filaggrin monomers depending on the species, e.g., 10 to 12 in humans, 16 to 20 in mice [15,16,17], and only 4 in dogs [18]. Profilaggrin is present in the stratum granulosum (SG), whereas filaggrin monomers are localized in the first layers of the SC. Of note, there is often confusion between profilaggrin and filaggrin, with some authors incorrectly assigning filaggrin to the SG. Filaggrin, together with other proteins of keratohyalin granules, forms the protein moiety of the cornified cell envelope of corneocytes [19,20,21,22,23]. In rat corneocytes, filaggrin was found to represent about 10% of the cell envelope-bound proteins [24]. Early work suggested the involvement of one or more specific phosphatases in filaggrin dephosphorylation [11,25], and in 1988, a rat 40 kDa phosphatase specific for profilaggrin was isolated [26] and later characterized as a protein phosphatase 2A-type [25]. However, data concerning specific profilaggrin phosphatases in humans are scant. 

In 1993, calpain 1 was proposed as a profilaggrin protease because profilaggrin cleavage requires calcium, and calpains are calcium-dependent neutral proteases [27]. In 2009, processing of human profilaggrin by human calpain 1 was confirmed in a cell-free proteolytic assay [28]. Moreover, cathepsin L-like proteinase, but not cathepsin E, isolated from rat epidermis was shown to hydrolyze profilaggrin [29]. Work in mice showed that loss of cathepsin H, produced by an shRNA approach, leads to a dramatic reduction of filaggrin monomers, owing to defective profilaggrin processing in the epidermis [30]. The results for cathepsin D are contradictory, and the enzyme is likely not essential for profilaggrin processing in vivo [29,31,32]. Furthermore, profilaggrin endoproteinase 1 (PEP1) has been identified as a cytoplasmic enzyme that is able to digest insoluble profilaggrin purified from mouse epidermis, at least in proteolytic assays [33]. In addition, both furin and PACE4, two calcium-dependent serine proteases belonging to the same family of proprotein convertases, can cleave profilaggrin in vitro at a site between the amino terminus and the first filaggrin repeat [34]. In mice, genetic deletion of matriptase/MT-SP1, a type II transmembrane serine protease expressed in epithelial cells, including KCs, leads to failure to process profilaggrin into filaggrin monomers [35]. Moreover, it has been shown that skin-specific retroviral-like aspartic protease (SASPase) activity is indispensable for processing profilaggrin in mouse epidermis [36], a requirement which may extend to human epidermis [37,38]. SASPase cleaves the linker sequence of human profilaggrin between ‘GSFLY’ and ‘QVSTH’ [36]. Moreover, it has been proposed that lympho-epithelial Kazal type inhibitor (LEKTI), a protease inhibitor encoded by the *SPINK*5 gene, controls profilaggrin proteolysis. Indeed, *SPINK5* knockout mice or mice with a premature stop codon in *SPINK5*—both are mouse models of Netherton syndrome, a severe ichthyosis due to null mutations in *SPINK5*—display increased amounts of mature filaggrin and reduced amounts of profilaggrin, indicating that proteolytic processing of profilaggrin is enhanced [39,40]. In human KCs, it was further shown that mesotrypsin liberates a 55-kDa N-terminal fragment of profilaggrin, hence potentially contributing to filaggrin processing as well [41]. Lastly, in transgenic mice overexpressing elastase 2, the rate of profilaggrin processing into filaggrin monomers is accelerated. In line with this, in the SG of patients with Netherton syndrome, elastase 2 is upregulated and co-localizes with profilaggrin, and profilaggrin levels are reduced [42,43,44]. Thus, many different proteases have been shown to potentially cleave profilaggrin into filaggrin monomers. 

In the upper SC, filaggrin is further processed by various proteases into free amino acids. The proteolysis of filaggrin is enhanced by its prior deamination or by citrullination. The latter is a post-translational modification catalyzed by peptidylarginine deiminase enzymes, resulting in the conversion of arginine into citrulline, thereby reducing its charge (citrulline is a neutral residue, whereas arginine is positively charged) and promoting detachment of the filaggrin monomer from the aggregated keratins [45,46,47]. Histidine and glutamine are then either enzymatically or spontaneously transformed to *trans*-urocanic acid (UCA) and pyrrolidone carboxylic acid (PCA), respectively. These amino acids and derivatives are components of the natural moisturizing factor (NMF), together with other molecules, such as lactate, chloride and sodium ions, and urea, thus ensuring proper SC hydration [48]. Interestingly, studies with histidinemic or caspase-14-deficient mice or with epidermal equivalents knocked-down for *FLG* have shown a role for *trans*-UCA in the protection of skin against the damaging effects of UV-B radiation [5,49,50,51,52,53,54].

In 1986, Scott and Harding established that proteolysis of filaggrin in rat skin is controlled by atmospheric humidity. Filaggrin is first detected at day 20 of rat gestation in all layers of the neo-formed SC. A few hours after birth, filaggrin disappears from the upper SC layers and accumulates in the innermost half of the SC, before being confined to a thin layer at the bottom of the SC, two days after birth. In addition, keeping the newborn rat in 100% humidity or applying occlusive patches onto the skin of adult rats prevents the activation of filaggrin degradation. The authors concluded that the lowering of atmospheric humidity promotes filaggrin proteolysis into water-retaining molecules [55]. This was confirmed first in mice [56] and then in humans, using 3D epidermal equivalents [57]. Indeed, when human epidermis is produced ex vivo at low atmospheric humidity, as compared to the typical saturated conditions, profilaggrin synthesis is increased, as is filaggrin degradation into UCA and PCA. This is in line with measurements conducted in humans, showing that NMF levels are low at birth and increase within hours or days after birth in healthy baby skin [58]. These results suggest that changes in the skin microenvironment, as occurs at birth, activate filaggrin degradation but not necessarily via increased protease activity. Indeed, it has been shown in vitro that reducing the external humidity increases filaggrin deimination and, as a consequence, the molecule’s dissociation from the filamentous corneocyte matrix [46,47,57,59]. This may render filaggrin more accessible to proteases [48]. Moreover, these results suggest that improper bundling of filaggrin with keratin filaments might promote filaggrin degradation in the absence of other alterations, such as increased activity of proteases. Besides external humidity, the neutral cysteine protease bleomycin hydrolase has been shown to participate in the breakdown of deiminated filaggrin into amino acids in a cell-free proteolytic assay [28]. Moreover, bleomycin hydrolase-deficient mice display reduced amounts of skin NMF and higher levels of filaggrin [60]. Furthermore, in healthy baby skin, regional upregulation of bleomycin hydrolase activity, e.g., in the cheek when compared to the elbow flexure, might contribute to increased skin hydration at specific body sites [58]. 

Thus, SASPase and calpain 1 might play important roles in the processing of profilaggrin into filaggrin, whereas bleomycin hydrolase and caspase 14 are probably major proteinases responsible for filaggrin breakdown in the skin (Table 1, Figure 1); however, the contribution of other proteinases cannot be ruled out. 

## 2. The Impact of the *FLG* Gene: From Ichthyosis Vulgaris to Atopic Dermatitis

Ichthyosis vulgaris (IV) is a common skin disorder in humans, characterized by dry, rough, and scaly skin. In 1985, it was first reported that keratohyalin granules, profilaggrin, and filaggrin are all reduced or absent in the epidermis of patients with IV and that these biochemical abnormalities correlate with the clinical severity of the disease [61]. In 1991, high-resolution ultrastructural immunolabeling showed many small filaggrin-positive granules in association with bundles of keratin filaments in IV SG [20]. However, unequivocally, the amounts of filaggrin are much lower in the SC of IV patients compared to that of healthy volunteers [20]. In 2006, genetic studies shed new light on the IV disorder, with the identification of loss-of-function mutations in the *FLG* gene associated with moderate or severe IV [62]. In the same year, Palmer et al. showed that *FLG* loss-of-function mutations are strongly associated with atopic dermatitis (AD) [63], suggesting that low amounts of filaggrin in AD might result from genetic causes. Indeed, earlier pioneering reports showed a decrease of filaggrin in lesional AD epidermis [64], regardless of skin lesion presence or absence [65,66]. AD, also known as atopic eczema, is the most common inflammatory skin diseases, affecting 1%–36% of children and up to 18% of adults, of whom approximately 20% have moderate-to-severe disease [67,68,69,70]. Here it is important to emphasize that reduced filaggrin levels are also observed in AD patients who are genotypically the wild type for *FLG* [65,71], demonstrating that *FLG* variants are not the only factors responsible for filaggrin downregulation in AD skin, and that other factors (e.g., environmental, metabolic) are also likely to be involved. Moreover, African American patients with AD exhibit normal levels of filaggrin, in contrast to Asian and European patients [66,71]. On the other hand, pediatric AD, regardless of the patient’s ethnic background, might be characterized by normal skin levels of filaggrin [72]; however, most studies lack either *FLG* genotyping or measurements of filaggrin amounts [64,72,73,74].

The half-life of rodent filaggrin in the SC is 6–9 h, demonstrating that filaggrin degradation is a relatively rapid process [16,75]. This high turnover of filaggrin may confer vulnerability to the epidermis because factors that lead to reduced filaggrin expression or that accelerate its degradation could have rapid effects on the biological processes in which filaggrin is involved. Conversely, environmental factors or therapies able to augment filaggrin expression might have rapid beneficial effects. Thus, it is important to decipher accurately all the physiological roles of filaggrin in the epidermis so that pathways potentially ameliorable by filaggrin-targeted therapy are fully identified.

## 3. *FLG* Null Mutations Are Strong Genetic Factors in AD

The strong association between *FLG* null mutations and AD (OR = 13.4), first observed by Palmer et al. [63], was subsequently confirmed by many other research groups in studies of various ethnic populations [76,77,78,79,80,81,82,83,84,85,86,87,88,89,90,91,92,93,94,95,96,97,98,99,100,101,102]. The strength of this association was initially reported to depend on *FLG* genotype; that is, in a cohort of 186 patients with AD, homozygosis for the combined null genotype of six major null mutations (R501X, 2282del4, R2447X, S3247X, 3702delG, and 3673delC) was more strongly associated with the disease (OR = 85.9) compared to heterozygosis (OR = 4.6) [103]. Furthermore, Brown et al. found, in a large cohort of children, that heterozygosis is not significantly associated with AD (OR = 1.2), as opposed to homozygosis (OR = 26.9) [104]. Nomura and Kabashima suggest that the initial overestimation of the strength of the association of *FLG* null mutations with AD, especially in heterozygous patients, is likely due to recruitment of patients with severe disease symptoms [67]. Nevertheless, it remains that *FLG* homozygosis or compound heterozygosis significantly increases the risk of developing AD, at least in adult European and Asian populations [71,105]. However, Morar et al. found *FLG* null mutations in 26.7% of young patients with AD, but also in 14.4% of children without AD [106]. Subsequent studies have confirmed that about 40%–50% of all carriers of *FLG* null alleles never experience eczema [107,108]. Thus, these data show strong but incomplete penetrance of *FLG* variants [63,80,109,110,111], whose impact on AD development might be modulated by ethnic-specific genetic modifiers, epigenetic alterations, or other environmental factors. 

Beyond *FLG* mutations, the number of filaggrin monomers encoded by an *FLG* allele can vary from 10 to 12 in humans. Individuals with 20 copies of the filaggrin monomer (i.e., homozygous 10/10) were reported to have an increased risk of developing AD, whereas those with 21 to 24 copies (genotypes 10/11 to 12/12) had no significantly elevated risk, at least in patients with moderate AD [112]. Similar work in patients with low-to-mild AD would give additional information on the power of the association between the different numbers of filaggrin monomers (i.e., genotypes) and the disease expression.

*FLG* null mutations have been shown to correlate with AD severity and persistence in adulthood [77,88,106,113,114]. Indeed, the observed higher prevalence of adult AD patients with an *FLG* null mutation (42%) [113] when compared to pediatric patients (13%) [104] argues for a role of *FLG* status in disease evolution, although this remains to be confirmed in large patient cohorts with different ethnic backgrounds. 

A new classification of AD patients according to age, ethnicity, concomitant atopy, inflammatory milieu, SC lipid abnormalities, and *FLG* status designated as ‘endotypes’ has recently been proposed. In this perspective, AD patients with *FLG* null mutations represent a particular endotype [66,115]. 

## 4. Decreased Filaggrin in Atopic Skin, Regardless of *FLG* Genotype

The relationship between *FLG* genotype and amounts of filaggrin in the epidermis is not a direct one, at least in AD. Howell et al. showed that filaggrin mRNA and protein levels are reduced in both non-lesional and lesional AD skin, regardless of *FLG* genotype [116]; this is a finding confirmed by others [65,117,118]. However, *FLG* heterozygosis further decreases filaggrin mRNA and protein amounts in acute skin lesions [116]. Thus, in AD skin lesions, *FLG* heterozygosis and the micromilieu might synergize to further diminish filaggrin amounts [116]. However, there may be differences between acute and chronic AD lesions, with a stronger reduction of profilaggrin compared to filaggrin in acute lesions and conversely in chronic lesions [65]. These differences likely result from modulation of profilaggrin or filaggrin processing by microenvironmental factors, because *FLG* mRNA levels are similar in acute and chronic lesions [119]. It has been hypothesized that immunological factors, notably Th2 cytokines, are responsible for modulation of filaggrin in AD skin. Indeed, KCs or human skin equivalents treated with various cytokines, i.e., IL-13, IL-4, IL-25, and thymic stromal lymphopoietin (TSLP) alone or in combination at doses ranging from 50 to 100 ng/mL, display reduced levels of filaggrin mRNA and protein [65,116,120,121]. Similar results have been found after treatment with IL-22, IL-17A, or IL-31 in the same dose range [122,123,124,125]; in contrast, IL-33 exerts only minor effects [126]. Data on IFN-γ are contradictory, even though all three IFN-γ studies utilized the same amount of cytokine, i.e., 20 ng/mL [116,120]. Thus, it is now well accepted that Th2 cytokines downregulate filaggrin. However, it must be emphasized that AD epidermis contains much lower levels of cytokines, i.e., in the picogram range in both human and mouse AD skin [127,128,129]. Therefore, the use of lower amounts of cytokines in in vitro experiments might better mimic the physiopathological skin microenvironment, including a potential synergetic effect of cytokines. 

Various studies in AD patients provide several hints about the effect of the skin cytokine milieu on filaggrin expression. *FLG* expression is normalized in AD patients treated with cyclosporine, concomitant with reductions in the expression of Th2, Th17, and Th22, which are key cytokines in the skin [130]. Moreover, *FLG* expression is also normalized in AD patients treated with Dupilumab, a monoclonal antibody that inhibits IL-4 and IL-13 signaling by binding to their receptors. This effect is associated with the downregulation of the expression of *IL17A*, *IL23*, and *IL22*, as well as the upregulation of several other cytokine genes, including *IL5*, *IL31*, and *IL33* [130,131]. The filaggrin mRNA level is further increased in the skin of high responders to Dupilumab treatment, in which *IL4*, *IL17*, *IL22*, *IL23A,* and *IL31* are all downregulated [130]. However, whereas Th2 inflammation is observed in all AD endotypes, filaggrin downregulation is endotype-specific [66], suggesting a marginal role of Th2 cytokines per se. Furthermore, an overarching role of Th22/Th17 inflammation is also not compatible with the modulation of filaggrin expression observed in the different AD endotypes [66]. Thus, it is likely that the inflammatory skin milieu alone is not sufficient to downregulate (directly or indirectly) filaggrin in AD patients who are genotypically the wild type for *FLG*. Other factors might significantly affect filaggrin expression in AD skin, as well. For example, the hypothesis of accelerated filaggrin proteolysis in AD, as mentioned above, has yet to be explored. 

The emphasis given to cytokines as the main effectors of filaggrin abnormalities in AD has practically sidelined thorough study of other potential regulators. Interestingly, sphingosyl-phosphorylcholine, which is upregulated in both lesional and non-lesional AD SC [132], has been shown to downregulate *FLG* mRNA levels in human KCs, presumably via upregulation of intracellular reactive oxygen species (ROS), NADPH oxidase 5, and cyclooxygenase-2 [133], suggesting that lipid abnormalities in AD might regulate filaggrin. In addition, mice with epidermal deletion of the *SIRT1* gene, encoding the Sirtuin1 protein, a lysine deacetylase, develop AD-like symptoms associated with nearly complete absence of profilaggrin and filaggrin [134]. In line with this, lesional AD skin displays low levels of Sirtuin1 [134]. However, the molecular mechanisms involved in sirtuin regulation of filaggrin expression are not clear [134]. Other pathways might dampen *FLG* expression in AD skin, such as the involvement of transcription factors (e.g., aryl hydrocarbon receptor gene (AHR), AP1, POU, and hypoxia-inducible factor (HIF)1/2α) that are known to bind the *FLG* gene promoter region; however, further work is required. Interestingly, mTOR/RAPTOR can downregulate *FLG* via attenuation of the AKT1 pathway [30]. Moreover, AKT1 has been shown to stabilize the binding of actin to profilaggrin in keratohyalin granules [135], potentially decreasing the hydrolysis of profilaggrin into monomers. This may be clinically important because AKT1 is markedly reduced in the skin of AD patients [30]. Interestingly, preliminary work on human KCs showed that epigenetic modifications, e.g., DNA methylation at the 5′-end of the *FLG* gene, might also modulate *FLG* expression [136,137], and DNA hypermethylation in the *FLG* promoter has been evidenced in lesional AD skin when compared to non-lesional AD skin [137]. However, in the latter study, the authors did not correlate the levels of DNA methylation and the expression of *FLG* in lesional versus non-lesional AD skin. Thus, additional regulators of *FLG* expression in AD, beyond the immunological microenvironment, deserve greater scrutiny.

The inflammatory microenvironment, skin dryness and/or dysbiosis, and/or abnormalities in the cytoskeleton might lead to de-coordination between *FL*G expression and filaggrin amounts in AD [36,138,139]. Pellerin et al. showed that, although the amounts of profilaggrin and filaggrin are both decreased in non-lesional and lesional skin of adult AD patients, regardless of *FLG* genotype, filaggrin is reduced more than profilaggrin [65,140]. This observation suggests that either (1) the activity of the proteases, which cleave profilaggrin into filaggrin monomers, is decreased or (2) the degradation of filaggrin into its breakdown products is increased in AD. The reduced amounts of activated caspase 14 and cathepsins and the reduced activity of bleomycin hydrolase observed in AD skin, irrespective of lesions [120,140,141,142], favor the first hypothesis (Figure 2). Of note, the rapamycin-insensitive complex mTOR2/AKT reduces profilaggrin processing into filaggrin in mouse epidermis, potentially via decreased mRNA levels of cathepsin H [143]. However, the role of mTOR2 in AD remains elusive [144]. 

Altogether, these data suggest that reduced levels of filaggrin in atopic skin, in addition to reductions caused by *FLG* nonsense mutations, might result from inhibition of signaling pathways in response to metabolic changes, inflammation, skin dysbiosis, or genetic polymorphisms, among other drivers. Moreover, the skin microenvironment might also control the level of methylation of *FLG* and, in turn, contribute to its downregulation in AD (Figure 3). Furthermore, the decreased ratio of filaggrin to profilaggrin in AD skin might result from reduced profilaggrin processing rather than from accelerated filaggrin degradation (Figure 2). Further studies are needed to identify the upstream signals responsible for profilaggrin catabolism in AD skin.

## 5. Filaggrin Deficiency Induces Subtle Epidermal Barrier Impairment in Non-Lesional AD

Several studies conducted in mice support a role of *FLG* loss-of-function mutations in causing SC leakiness to environmental molecules, i.e., in lessening the outside-in barrier. The first reports were carried out in flaky tail mice, which exhibit a frameshift mutation in *Flg*, leading to the synthesis of a truncated profilaggrin. These mice show increased paracellular permeability barrier of the SC to hydrophilic molecules—water-soluble molecules are normally not able to penetrate the SC due to its high lipid content [149,150,151]. Unfortunately, flaky tail mice also carry a spontaneous mutation in the Transmembrane Protein 79/Matt gene (*Tmem79*) and exhibit AD-like dermatitis, which precludes conclusions on the role of filaggrin based on this model. Thus, to circumvent off-target effects due to this additional mutation and to skin inflammation, *Flg* knockout mice have been utilized. Topical application of various molecules (unmodified or liposome-encapsulated) onto the skin of *Flg* knockout mice demonstrated increased SC permeability [6]. Moreover, topical application of either croton oil, hapten, or ovalbumin onto the skin of these mice resulted in increased skin inflammation, edema, and serum IgE [6], suggesting an altered outside-in barrier able to induce local and systemic inflammation. However, mouse epidermis, including the SC, is much thinner than human epidermis, leaving unanswered the real capacity of *FLG* null mutations to increase the permeability of human SC. Nevertheless, in biopsies from IV patients, lanthanum nitrate has been shown to progress via paracellular pathways into and across the SC, whereas there is no such movement in the skin of healthy volunteers [7]. Furthermore, *FLG* homozygous/compound heterozygous IV patients exhibit altered corneodesmosome structure and abnormal topography of the corneocyte surface [7,60]. Moreover, morphological changes of corneocytes and of corneodesmosomes have been observed in IV, likely resulting from retraction of keratin bundles and abnormal lamellar body secretion, respectively [7,152]. Previous work showed that kallikrein 7, a serine-protease able to cleave corneodesmosin [153], is upregulated in *FLG* knockdown human epidermal equivalents and the skin of AD patients [154,155]. Thus, filaggrin deficiency might alter corneocyte and corneodesmosome morphology enough to increase the permeability of the human SC to environmental compounds (outside-in barrier). However, further work is required to validate this hypothesis. Indeed, *FLG* null mutations do not increase the diffusivity of PEG 370 through the non-lesional skin of AD patients [156]. Furthermore, in *FLG* knockdown full-skin organotypic models, there were no changes in the flux of a lipophilic model compound, butyl-PABA, through the SC, clearly demonstrating that filaggrin deficiency does not affect SC permeability to this molecule [157]. These data highlight the difficulty in verifying the hypothesis of increased SC leakiness in the context of filaggrin deficiency and in performing functional assays with human skin. Indeed, a proper analysis of SC permeability requires multiple assays, including permeability and diffusion studies (e.g., Franz cell chamber) and experimental models [158].

*Flg* knockout mice do not exhibit increased transepidermal water loss (TEWL) [6]. This is in line with unaltered SC lipid organization, lateral packing, and content (ceramides; free fatty acids, including (very)-long-chain fatty acids; cholesterol) observed in *FLG* knockdown full-skin organotypic models [154] and in *Flg* knockout mice [156]. Accordingly, *FLG* knockdown does not alter the expression of enzymes involved in free fatty acid elongation, such as Elongation of very-long chain fatty acids (ELOVL)1, 4, and 6; and of stearoyl-CoA desaturase 1, β-glucocerobrosidase, and α-sphingomyelinase in both epidermal equivalents [125] and mouse skin [159]. Furthermore, the absence of epidermal tight junction morphological changes and leakiness in *Flg* knockout mice demonstrate that filaggrin deficiency has no deleterious effect on the paracellular barrier in living epidermal layers [160]. Thus, filaggrin deficiency per se does not alter the epidermal inside-out barrier [152,161].

In the context of AD, *FLG* null mutations have little or no impact on the epidermal barrier healing process, as evidenced by the lack of effects on SC barrier recovery after tape stripping or after skin irritation [162,163]. Moreover, as emphasized by Jung and Stingl, overt morphological and functional defects in the epidermal barrier in AD are independent of *FLG* null mutations [164,165]. In line with this, TEWL in non-lesional AD skin is similar between *FLG*-mutated patients and *FLG* wild-type patients [156,162,166,167,168,169], suggesting no additional impairment of the epidermal barrier in AD skin due to *FLG* loss-of-function mutations [170]. These results are corroborated by similar relative compositions and amounts of SC ceramides—a profile which directly correlates with epidermal barrier efficacy [171,172,173]—in non-lesional skin of *FLG*-mutated and *FLG* wild-type AD patients [162,167]. 

Thus, filaggrin deficiency, regardless of genotype, contributes to an abnormal lamellar body cargo system and secretion and alters corneocyte morphology, resilience, and surface properties by acting on keratin organization and cornified envelopes. Moreover, filaggrin deficiency alters the ultrastructure of corneodesmosomes, as well as the SC lipid matrix (i.e., disorganized lamellar bilayers with incompletely processed lamellar material and lacunae). All of these subtle abnormalities of the SC might favor the penetration by specific but not all molecules according to their physicochemical properties. 

## 6. Filaggrin Deficiency Promotes Subclinical Inflammation in the Epidermis Nonspecific to AD

Proteomic analysis of *FLG* knockdown in human epidermal equivalents showed increased amounts of inflammation-related proteins, albeit with only weak or indirect links to specific inflammatory processes [153]. In contrast, *FLG* knockdown in human differentiated KCs was shown to augment the production of various cytokines involved in barrier repair, including IL-1α, GM-CSF, IL-8, and IL-18 [174]. However, in epidermal equivalents generated with primary KCs from IV patients, the expression of cytokines involved in barrier repair and AD-specific inflammation was not increased [152]. Discrepancies between the phenotypes of *FLG* knockdown and primary IV KCs might be due to temporal effects, i.e. gene knockdown is a relatively quick process during which cells do not have time to compensate the protein loss, in contrast to the situation in patient cells [161,175,176]. 

TSLP has been shown to be increased in skin lesions of AD patients and to play an important role in AD pathogenesis via activation of Langerhans cells, at least in mouse models [177,178,179,180,181]. Wallmeyer et al. found that *FLG* knockdown in human full-skin organotypic models results in increased TSLP [182]. However, other work carried out in epidermal equivalents generated with IV patient KCs did not show increased *TSLP* expression [152], similar to *Flg* knockout mice [183], suggesting a lack of significant effects of filaggrin deficiency per se on *TSLP* expression [184]. In AD patients with *FLG* null mutations, Langerhans cell maturation is promoted, a complex process potentially including reduced amounts of UCA [185]. Thus, filaggrin deficiency in vivo might lead to subclinical skin inflammation via Langerhans cell activation rather than via upregulation of pro-inflammatory cytokines in KCs. 

Work undertaken in mice showed that filaggrin deficiency promotes AD-like symptoms in mice with a BALB/c background [183], in contrast to mice with a C57BL/6 background [186], showing that (1) filaggrin is a permissive but not an eliciting factor; and (2) other acquired factors, including immunological susceptibility, are important in AD pathogenesis. Indeed, filaggrin deficiency might promote a Th17 immune response [187,188,189] that can be found not only in AD but also in other inflammatory skin diseases (e.g., psoriasis and ichthyoses). Interestingly, filaggrin is expressed in the mouse thymic medulla [190], suggesting further systemic immunomodulatory effects of filaggrin deficiency. In line with this, proportions of circulating Th2- or Th17-regulatory T cells (Tregs) are increased in AD and accentuated by *FLG* null mutations. Moreover, *FLG* null mutations are associated with expansion of thymus-emigrated Tregs in AD [188].

Thus, filaggrin deficiency does not trigger inflammatory pathways specific to AD but might contribute to Th17 inflammation via regulation of T-cell priming in the thymus and to Langerhans cell activation via, inter alia, reduced levels of UCA, whose role in immunological reactions in the skin deserves further investigation [191]. However, most work has been carried out in mice or in vitro and warrants validation in patients.

## 7. Does Filaggrin Deficiency in Atopic Skin Lead to Alkalization of the Skin?

Free fatty acids, components of the NMF (including UCA), and the Na/H proton pump all are believed to contribute to the establishment of an acidic skin surface pH, thereby limiting the overgrowth of microbes [192] and controlling desquamation through protease activity. Thus, filaggrin is believed to contribute to surface-skin pH acidification via its degradation into UCA. The first evidence of a role for UCA in skin-surface pH was published in 2000 [193]. To experimentally identify the main biologic components involved in the acidity of the SC, the authors placed a drop of water in contact with the skin for 30 min to collect proton donors. Then, using a mathematical extrapolation model, they showed a significant contribution of UCA to surface-skin acidification [193]. However, the use of water as a solvent prevents the collection of water-insoluble proton donors, such as fatty acids, whose pKa can be similar to or even higher than that of UCA (e.g., pKa stearic acid = 10.15 versus pKa UCA = 6.1) [193,194]. Of note, the ratio of free fatty acids to UCA is about 6–7 in human epidermis. Thus, any significant contribution of UCA to acidification of the skin surface can be questioned. Moreover, several studies suggest that filaggrin deficiency does not significantly alter surface-skin pH. *Flg* knockout mice do not display significant changes in surface-skin pH [6]. Moreover, studies in IV patients showed either unchanged [163] or slightly increased skin-surface pH but still within the acidic range (≤6) [7,142,170,195]. In contrast, it has been observed that a decreased amount of UCA parallels an increased surface pH in human epidermal equivalents knocked-down for filaggrin 2, in which profilaggrin processing is impaired and there is a reduced amount of filaggrin [196].

Thus, the role of filaggrin in acidification of the skin surface via the production of UCA can still be debated [197]. As a corollary, increased protease activity in lesional AD skin [198,199] is an unlikely outcome of filaggrin deficiency. It is more likely that filaggrin breakdown-products significantly contribute to water retention and skin osmolarity, and thereby to SC flexibility [60]. Moreover, it is often forgotten that skin bacteria can use histidine as a substrate and be a significant local source of UCA [200]. 

## 8. Effects of Filaggrin Deficiency on the Skin Microbiota in AD 

Several lines of evidence suggest a role for filaggrin deficiency in controlling the composition of the skin microbiota. Filaggrin-deficient individuals, regardless of AD, harbor an altered bacterial skin microbiota, with reduced proportions of Gram-positive bacteria, namely Finegoldia (*Finegoldia magna*), Anaerococcus, and Peptoniphilus, which are all capable of using histidine as a nutrient (carbon) source. This might be a consequence of nutrient competition driven by the loss of histidine-rich filaggrin [195]. However, *Finegoldia magna* utilizes amino acids such as glycine, serine, and threonine as major energy sources [201,202], and Anaerococcus and Peptoniphilus primarily use peptone [203]. Because filaggrin is rich in glycine (13.6% of total amino acids) and serine (25.3%), the reduced proportions of Gram-positive bacteria on the skin of filaggrin-deficient individuals might also be related to reduced levels of glycine and serine in the SC. 

Frequent microbial skin infection in AD patients seems unlikely to result simply from the observed small shift in surface-skin pH, which remains in the acidic range in AD patients, including in severe skin lesions, regardless of *FLG* genotype (see below). However, filaggrin deficiency, irrespective of genetic or environmental origins, might contribute to skin dysbiosis and its various clinical manifestations by reducing the levels of UCA and PCA, substances which have been shown to exert antimicrobial properties [189]. Indirect evidence comes from work on caspase-14-deficient mice, which showed impaired clearance of *Escherichia coli* after topical challenge [204]. These results suggest a role for filaggrin breakdown products in the clearance of skin pathogens. 

*FLG* null mutations have been associated with modification of the microbiota more in non-lesional AD skin than in lesional AD skin [205], notably by decreasing the relative abundance of Proteobacteria and increasing that of Actinobacteria [206]. Moreover, there is an inverse correlation between *Staphylococcus aureus* and *Cutinobacterium acnes* in AD skin [207], suggesting that these bacteria are competitors or that the skin microenvironment favoring the growth of one of these bacterial species is strongly unfavorable to the growth of the other [207]. Accordingly, in IV and AD patients without *Staphylococcus aureus*, the amount of *Cutinobacterium acnes* is increased when compared to healthy controls [208]. Moreover, the amount of *Cutinobacterium acnes* is increased in *FLG*-mutated AD patients when compared to *FLG* wild-type AD patients [208], suggesting that physicochemical changes to the skin microenvironment induced by filaggrin deficiency might promote the growth of *Cutinobacterium acnes*. However, no associations have been found between acne and *FLG* null mutations [209]. Thus, *FLG* null mutations might favor the growth of *Cutinobacterium acnes* to an extent that allows bacterial competition but not to a level that provokes acne (i.e., dysbiosis, not infection). 

*FLG* status might modulate the skin fungal microbiota in AD, as IV patients exhibit fungal dysbiosis [208] and increased fungal skin infections [210]. However, little work has been carried out in this direction so far. A pilot study showed that *FLG* null mutations are modifiers of the fungal microbiota in AD, notably by leading to enrichment in *Cladosporium* and *Malassezia*, especially *Malassezia globosa*, and to diminution of *Debaryomyces* and *Leptosphaeria* [208]. Thus, *FLG* null mutations might accentuate fungal dysbiosis in AD, which might steepen Th17 immunity [187,188,189]. 

*FLG* null mutations confer increased risk to eczema herperticum (OR = 3.4) in both European and African ancestry populations [211]. However, the role of filaggrin deficiency in susceptibility to viral infection in AD patients remains almost uninvestigated. 

## 9. Filaggrin Deficiency per se Does Not Promote Colonization of Atopic Skin with *Staphylococcus aureus*


A recent pilot study found that, while patients with IV display increased proportions of staphylococcal species on their skin, *Staphylococcus aureus* is not detected [208]. Absence of *Staphylococcus aureus* was also reported in an earlier study of IV [195]. Thus, initial evidence does not support a strong association between filaggrin deficiency per se and skin colonization with *Staphylococcus aureus*; however, further work in larger IV patient cohorts is required.

In the context of AD, the data concerning the status of *Staphylococcus aureus* lack clarity. Recent work has shown that, in lesional AD skin, *Staphylococcus aureus* [212,213] and especially the CC1 clonal lineage [212] are more prevalent in *FLG*-mutated than in *FLG* wild-type AD patients. The type of genetic polymorphism (nonsense vs. missense) in *FLG* might be important for colonization of the skin by this bacterium in AD [213]. Conversely, Indian patients with moderate-to-severe AD and a higher number of loss-of-function *FLG* alleles display a lower abundance of *Staphylococcus aureus* [214]. Moreover, in a German cohort of AD patients with moderate-to-severe AD, no specific enrichment of the skin microbiota with *Staphylococcus aureus* was found in *FLG*-mutated patients [206], similar to findings in Danish and Austrian cohorts with mainly low-to-moderate AD [205,208].

A current hypothesis states that the increased skin-surface pH observed in AD is a consequence of reduced NMF levels (due to reduced filaggrin proteolysis or *FLG* haploinsufficiency), which, in turn, favors skin superinfection with *Staphylococcus aureus* [215]. It is assumed that the bacterial superinfection then damages the epidermal barrier and elicits an immune response in KCs. However, this hypothesis is challenged by several observations. First, *Staphylococcus aureus* is present in skin lesions of AD patients with moderate-to-severe AD, whereas it is rare or absent in AD patients with low-to-mild symptoms, regardless of *FLG* status [195,205,206,207,208,216,217,218]. Second, skin-surface pH is not consistently increased in AD, but when it is increased, it remains in the acidic range (≤6), including in AD skin lesions and regardless of *FLG* status [142,167,215,219,220]. Moreover, *Staphylococcus aureus* can proliferate over a pH range from 5 to 9, with no major disruption to growth between pH 5.5 and 7.0 [221]. Thus, the minor variation of skin-surface pH observed in AD cannot explain *Staphylococcus aureus* overgrowth [206]. Third, filaggrin does not play a major role in the establishment of acidic skin-surface pH, as already discussed above. Of additional note, although several studies in vitro have shown that staphylococcal enterotoxin B (SEB), lipoteichoic acid, or various proteases secreted by *Staphylococcus aureus* are able to damage the epidermal barrier and induce an immune response in KCs [222], in vitro treatment of KCs or organotypic cultures with these molecules is only a partial experimental approach to the situation in vivo. 

Van Drongelen et al. showed that, in *FLG* knockdown human epidermal equivalents infected with *Staphylococcus aureus*, the proportions of adherent bacteria are increased, suggesting that filaggrin deficiency might promote the attachment of the bacteria to the SC. The authors extrapolated, in the context of AD, that *FLG* null mutations might promote skin superinfection via this mechanism [125]. Because patients with IV exhibit skin dysbiosis toward increased staphylococci and specifically toward increased *Staphylococcus epidermidis*, whereas *Staphylococcus aureus* is absent in the skin of these patients [208], it would be interesting to repeat the experiments of Van Drongelen et al. with *Staphylococcus epidermidis*. Although previous work has shown that filaggrin deficiency does not increase the risk of skin superinfection with *Staphylococcus aureus* in AD [206,208,223], it still might be an aggravating factor. In 1986, *Staphylococcus aureus* toxin was shown to bind to filaggrin [224], suggesting that the absence or strong reduction of filaggrin might facilitate alteration of the epidermis by preventing the retention of the staphylococcal toxin within the SC. Moreover, filaggrin might protect KCs from the toxicity of staphylococcal α-toxin by secreting sphingomyelinase [225]. 

Thus, filaggrin deficiency per se might not promote colonization of atopic skin with *Staphylococcus aureus* and, a fortiori, not via increased skin surface pH (as discussed above). It is more likely that an attenuated innate immunity, owing to impairment of the epidermal barrier [226], lifts the growth inhibition on *Staphylococcus aureus* in lesional AD. However, in superinfected skin, filaggrin deficiency might favor *Staphylococcus aureus* binding to the SC [224] and enhance the deleterious effects of staphylococcal toxin on the SC [227,228]. Interestingly, low levels of NMF strengthen *Staphylococcus aureus* binding to corneocytes via an interaction between the bacterial adhesion clumping factor B (ClfB) and corneocyte surface ligands or the N-terminal region of corneodesmosin [227,228].

## 10. Filaggrin Deficiency Leads to Cellular Abnormalities in Keratinocytes: Potential Relevance in AD

In AD, *FLG* null mutations worsen the cellular and molecular abnormalities in KCs [142]. Microarray analysis of skin initially showed that several pathways, such as focal adhesion, ECM receptor interaction, and regulation of the actin cytoskeleton, are more altered in *FLG*-mutated than in *FLG* wild-type AD patients [142]. Moreover, other cellular features, such as the phosphatidylinositol signaling system, calcium signaling, T-cell receptor signaling, ABC transporters, and tight junctions, are significantly altered only in *FLG*-mutated AD patients when compared to wild types [142]. Analysis of the crystal structure of specific domains of human profilaggrin demonstrated that it might bind to three proteins, namely annexin II/p36, stratifin/14-3-3 sigma, and heat shock protein 27 [229], which are involved in cell motility, linkage of membrane-associated protein complexes to the actin cytoskeleton, endocytosis/exocytosis, ion channel formation, cell matrix interactions, KC differentiation, and apoptosis. Thus, it would be interesting to see whether these interactions do take place in KCs and whether filaggrin deficiency could affect processes controlled by these binding partners.

In 1997, Dale et al. transiently transfected rodent epithelial cells with constructs encoding one-to-five human filaggrin monomers and observed cell contraction, nuclear membrane breakdown, and nuclear condensation, which are hallmarks of cell apoptosis. They proposed that filaggrin might be involved in KC death in the upper SG [4]. However, in 2000, the same group reported that inducible overexpression of mature filaggrin in KC cell lines did not increase cell death. Nonetheless, they showed that filaggrin-positive cells were more susceptible to cell death after treatment with UV-B or hydrogen peroxide [230]. The authors speculated that KC apoptosis at the transition zone between the SG and SC, which is a process requiring oxidative stress, might be facilitated by filaggrin [230,231].

The amino-terminal sequence of profilaggrin exhibits significant homology to the so-called small calcium-binding proteins such as S100 [232]. Because the S100 domain can bind calcium, profilaggrin has been proposed to play an important role in KC differentiation, potentially via activation of calcium-dependent protein kinase C (PKC) [232], which has been shown to induce KC differentiation [233]. It was shown that the N-terminal AB fragment of profilaggrin comprising the S100 domain and a nuclear localization signal can be translocated into the nucleus of KCs [41,234], and its accumulation there induces DNA degradation and cell death [41], diminishes cell proliferation, and alters differentiation [234]. Mesotrypsin liberates a 55 kDa N-terminal fragment of profilaggrin able to migrate into the KC nucleus [41]. Similarly, caspase 14 increases the production of the filaggrin AB fragment, hence potentially contributing to KC death [41]. Thus, several pieces of evidence show that cleavage of profilaggrin in granular transient KCs might release a protein fragment that is able to migrate into the nucleus and contribute to cell death. Thus, profilaggrin processing might significantly participate in the cornification process. However, it should be noted that IV patients exhibit degenerating nuclei in granular KCs, suggesting an impaired process of nucleophagy potentially leading to premature cell death [62]. Several lines of evidence suggest that, in AD and IV, absence or reduced amounts of profilaggrin might impair proper KC terminal differentiation rather than contribute to KC hyperproliferation [152,157,229,235].

Thus, in AD, low amounts of profilaggrin (not filaggrin) might alter the cornification process in KCs. This might also perturb cell metabolism, potentially inducing premature cell death via nucleus degeneration, thereby preventing the proper expression of other proteins of the cornified envelope such as hornerin, involucrin, and loricrin [236,237]. However, further work is required to validate or dismiss this hypothesis.

## 11. *FLG* Null Mutations Regulate Skin Moisture in Non-Lesional AD

Heterozygosity for *FLG* null mutations augments the risk of dry skin in both women and men (OR = 2.7–7.4) [238,239], and wild-type individuals with 10 filaggrin repeats (10/10 genotype) have drier skin (self-perceived) than those with 12 repeats (12/12 genotype) [240], including in AD [166,241]. The amount of NMF is reduced in ichthyotic individuals carrying *FLG* null mutations [168,242,243,244] and in *FLG*-mutated AD patients [241]. In AD skin, the concentrations of filaggrin breakdown products correlate with *FLG* copy number, with UCA having a greater correlation than histidine or PCA [112]. Work in mice suggests that low levels of NMF resulting from filaggrin deficiency confer stiffness to corneocytes [60], and this might ultimately lead to reduced skin elasticity and promote itch. However, recent work by Ota et al. showed that the development of skin inflammation during the transition between non-lesional to lesional AD correlates with reduced levels of filaggrin breakdown products, regardless of *FLG* null mutations [245].

Observations in IV patients demonstrate that *FLG* null mutations induce dry skin, indicating that the amount of filaggrin in the skin is a determinant of proper skin moistness. In non-lesional AD, *FLG* null mutations leading to reduced filaggrin result in dry skin, whereas non-lesional AD skin from *FLG* wild-type patients does not exhibit xerosis. In contrast, in lesional AD, the skin is dry in all AD patients, irrespective of *FLG* null mutations, showing that other factors, such as metabolic, inflammatory, and/or microbial microenvironments, supersede the *FLG* genotype.

## 12. *FLG* Null Mutations Are Less Penetrant Genetic Factors in the Atopic March than in AD

The atopic march describes the progression across time of AD to allergic rhinoconjunctivitis and asthma. However, this evolution does not affect all AD patients. Indeed, it mainly depends on disease severity, since about 70% of patients with severe AD develop asthma versus 20%–30% of patients with mild AD. Similar observations have been made for allergic rhinoconjunctivitis, whose risk in AD patients correlates with the levels of total and specific IgE antibodies [246]. Initial work by Palmer et al. showed that *FLG* null mutations are significantly associated with asthma (OR = 1.8), although to a much lesser extent than with AD (OR = 13.4) [63], thus suggesting a role for *FLG* null mutations in the atopic march [247,248,249,250]. Indeed, it is now clear that *FLG* null mutations are associated with asthma only in the context of AD [251,252,253], because this association is lost in large patient cohorts of all ages without AD [104,254]. Moreover, the resolution of atopic symptoms in elderly individuals, despite *FLG* null mutations [114,255], suggests that the immune system overrides any genetic predisposition with regard to atopy.

Filaggrin is strongly expressed in the cornified epithelium of the nasal vestibular lining, but not in the transitional and respiratory nasal epithelia, which might explain why *FLG* null mutations are a risk factor for allergic rhinoconjunctivitis (OR = 2.64, penetrance 15.4%) [251]. Weidinger et al., the authors of that study, suggest that it is unlikely that *FLG* null mutations exert organ-specific and local effects in the upper airways; rather, they privilege a mechanism through which the mutations might contribute to increased percutaneous priming and, in turn, to the induction of Th2 inflammation in nasal epithelium [251].

The higher levels of serum IgE in AD patients with *FLG* null mutations initially suggested a role for filaggrin in allergic sensitization [77,106]. Subsequently, this association was confirmed in several studies [251,253,256,257,258,259,260]. The hypothesis is that filaggrin deficiency renders the SC more permeable to environmental allergens. In line with this, a strong association between *FLG* null mutations and sensitization to grass, house dust mites, cat dander, and other allergens has been evidenced in various ethnic populations [107,166,261]. In two independent cohorts, a significant interaction between *FLG* loss-of-function mutations and cat (but not dog) ownership at birth on the development of early-life eczema has been demonstrated [262]. Moreover, AD patients with *FLG* null mutations display increased blood Derp1^+^ CD4^+^ T cells (Derp1 is a house dust mite allergen), hence providing molecular immunological evidence of a link between *FLG* null mutations and systemic T-cell reactivity to allergens [263].

Increased susceptibility to peanut allergy has been consistently found in *FLG* carriers (OR = 3.0–5.4), suggesting increased sensitization via SC leakiness, as traces of peanuts were found in dust deposited on a lounge-sofa [235,258,264,265,266]. Interestingly, in a recent study, Berdyshev et al. showed that lower levels of filaggrin breakdown products, namely UCA and PCA, correlate with peanut allergy, irrespective of AD [267], suggesting that inflammation is not involved in the increased permeability of the skin to peanut allergens. Unfortunately, the authors did not genotype the study participants, hence precluding a potential association with *FLG* genotype. Thus, the relationship between *FLG* null mutations and food allergies deserves further work [268,269]. 

A controversy over the association between allergic dermatitis and *FL*G null mutations exists [239,265,269,270,271,272,273,274]. For example, there is no association between latex allergy and *FLG* null mutations [275,276]. Moreover, an allergen avoidance strategy as a measure of prevention both during pregnancy and after childbirth yielded disappointing results in AD patients, regardless of *FL*G status [269]. 

An association between *FLG* null mutations and allergic contact dermatitis to nickel (OR = 1.74–4.04) has been consistently observed in several studies [272,277,278,279,280,281]. Moreover, *FLG* null mutations seem to lower the age of first appearance of nickel allergy and the threshold to nickel reaction [279]. Interestingly, filaggrin can strongly bind to nickel in the human epidermis [282]. In addition, filaggrin was shown to be susceptible to nickel-assisted peptide bond hydrolysis, hence generating abnormal filaggrin fragments [283]. These results suggest that the binding of nickel to filaggrin in the SC might retain nickel and prevent its further progress into the epidermis to elicit an allergic reaction. Thus, in this context, reduced filaggrin might favor skin sensitization by rendering the SC more permeable to nickel. An exciting question is whether filaggrin might exert similar retention properties for other allergens, especially where the structures of specific allergens are compatible with strong binding to filaggrin. It is noteworthy that filaggrin has been shown to bind to several proteins (https://varsome.com/gene/FLG, first accessed on 13 April 2022).

## 13. Conclusions

Since the discovery of the strong association between *FLG* null mutations and AD fifteen years ago, the role of filaggrin in AD pathogenesis has been extensively investigated [284,285]. It is now well established that profilaggrin and filaggrin have at least three critical roles in KCs: (1) alignment of keratin intermediate filaments; (2) control of cell shape; and (3) maintenance of epidermal texture via production of water-retaining molecules, namely UCA and PCA (Figure 3). Moreover, there is strong evidence that UCA is also involved in protection against solar radiation. Furthermore, filaggrin deficiency alters the ultrastructure of corneocytes and the SC lipid bilayer to facilitate the ingress of exogenous molecules, depending on their physicochemical properties, into living epidermal layers. The capacity of filaggrin to bind foreign molecules, including allergens and bacterial toxins and, in turn, prevent their penetration into the skin deserves further investigation, as does the role of filaggrin in the cornification process by promoting granular KC death at the SG–SC transition. However, the contribution of *FLG* null mutations to surface-skin pH and to excessive TEWL in AD is probably marginal. Finally, sifting the wheat from the chaff with regard to filaggrin functions in skin homeostasis and in AD pathogenesis will not only advance current basic knowledge in dermatology but will also help deliver reliable data to the skin research community, as such data are indispensable to the design of effective therapies for patients with *FLG* null mutations.

## Figures and Tables

**Figure 1 ijms-23-05318-f001:**
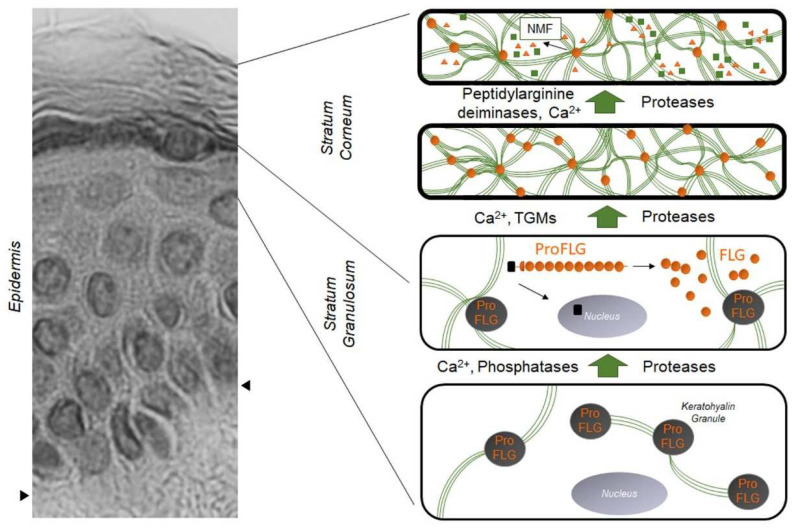
Evolution and distribution of profilaggrin and filaggrin during epidermal differentiation: dark gray disks containing profilaggrin are keratohyalin granules present in the cytoplasm of granular KCs. Under the action of phosphatases, proteases, and Ca^2+^, profilaggrin is expelled in the cytoplasm and degraded into filaggrin monomers (small orange disks). Green lines are keratin filaments aggregated by filaggrin monomers in the lower SC. Under the action of transglutaminases (TGMs), filaggrin molecules may be covalently linked to the cornified envelope. In the upper SC, filaggrin is further processed into free amino acids by the sequential action of peptidylarginine deaminases and proteases, producing the Natural Moisturizing Factor (NMF, green squares and orange triangles). The left part of the figure is a hematoxylin–eosin staining of a healthy human epidermis; arrows show the dermis–epidermis junction.

**Figure 2 ijms-23-05318-f002:**
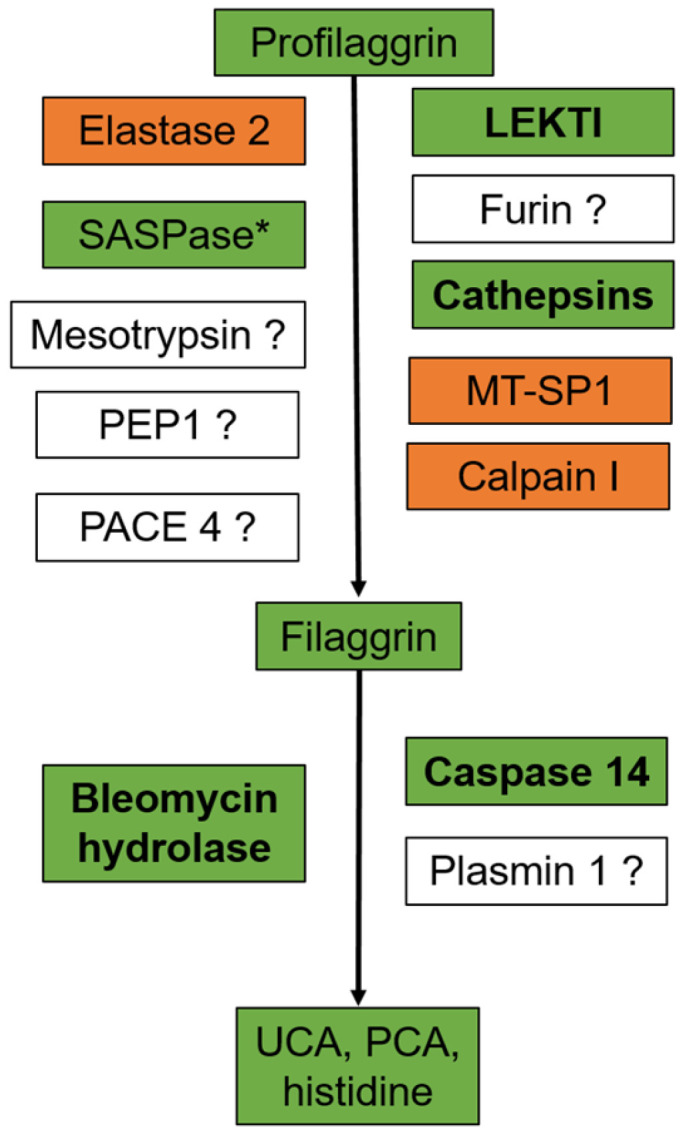
Control of profilaggrin and filaggrin processing in atopic dermatitis: summary of data collected in mouse models and patients with atopic dermatitis. Changes in protein amounts in atopic skin is shown by using a color code (orange, increase; green, decrease). Enzymes written in bold show regulation consistent with modulation of amounts of profilaggrin and filaggrin observed in atopic skin [30,120,140,141,145,146,147,148]. * Only studied in mice. PEP1, profilaggrin endoproteinase 1; LEKTI, lympho-epithelial Kazal-type-related inhibitor; MT-SP1, matriptase; UCA, urocanic acid; PCA, pyrrolidone carboxylic acid.

**Figure 3 ijms-23-05318-f003:**
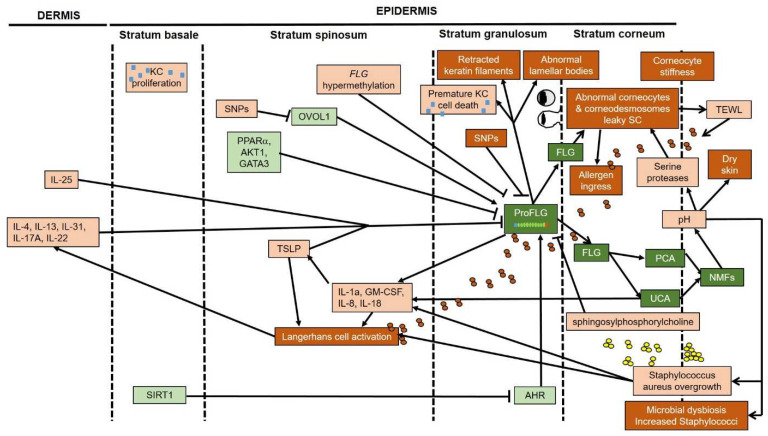
Summary of filaggrin regulation and roles in atopic dermatitis: experimentally confirmed roles (darkest boxes, white writing) versus prevailing ideas, most of them unsubstantiated. Decreased parameters are in green boxes, and increased parameters are in orange boxes.

**Table 1 ijms-23-05318-t001:** Summary of enzymes and enzyme inhibitors involved in filaggrin metabolism (sometimes still hypothetic) in the epidermis according to data generated in vitro and in animal models versus in humans.

	In Vitro and Animal Models	in Humans
Dephosphorylation of profilaggrin	Phosphatase of the protein phosphatase 2A family	
Profilaggrin processing	Calpain 1, cathepsins L-like and H, PEP1, furin, PACE4, MT-SP1, mesotrypsin, SASPase, LEKTI, elastase 2	Calpain I, SASPase, mesotrypsin
Filaggrin citrullination in the stratum corneum	PAD1 and/or 3	PAD1 and/or 3
Proteolysis of filaggrin in the stratum corneum	Bleomycin hydrolase, calpain 1, caspase 14	Bleomycin hydrolase, caspase 14

## Data Availability

Not applicable.

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
