# Peer review of "Revisiting the Roles of Filaggrin in Atopic Dermatitis"

_ijms, 2022, doi:10.3390/ijms23105318_

Round 1
Reviewer 1 Report
The authors summarized the history, possible contribution, and unanswered questions of filaggrin in atopic dermatitis really well. The two major comments below would make the manuscript informative and easy to be understood.
Major comments:
1. Please make a table that summarize the enzymes that cleave filaggrin.
2. Lines 188-194: I could not understand what “copy number” of filaggrin means. Please explain it in an easy way.
Author Response
ijms-1706956
We would like to thank the reviewer for helping us to improve this review on the roles of filaggrin in atopic dermatitis. We have amended the manuscript according to suggestions. Moreover, the language has been revised.
Reviewer:
The authors summarized the history, possible contribution, and unanswered questions of filaggrin in atopic dermatitis really well. The two major comments below would make the manuscript informative and easy to be understood.
Response: First of all, we would like to warmly thank the reviewer for this nice comment. We take it very gladly.
Major comments:
Please make a table that summarize the enzymes that cleave filaggrin.
Response: as suggested by the reviewer, we have added a table (table 1) detailing the enzymes that cleave filaggrin.
Lines 188-194: I could not understand what “copy number” of filaggrin means. Please explain it in an easy way.
Response: We have amended the manuscript to render the sentence easier to understand. Modifications asked by the reviewer have been underlined in the revised text.
Reviewer 2 Report
A very complete narrative review on filagrin, from its discovery ot its currrent implications in dermatology and atopic dermatitis; i really enjoyed reading it; only minor revisions:
In the text, the acronim for atopic dermatitis is not reported, just in the abstract...please check.
Line 150, you should add: "Atopic dermatitis (AD), also known as atopic eczema, is one of the most common inflammatory skin diseases, affecting
15%-30% of children and up to 14.3% of adults, of whom
approximately 20% have moderate-to-severe disease" and cite : doi: 10.18176/jiaci.0519. and doi: 10.1111/exd.14276.
Thank you
Author Response
ijms-1706956
We would like to thank the reviewer for helping us to improve this review on the roles of filaggrin in atopic dermatitis. We have amended the manuscript according to suggestions. Moreover, the language has been revised.
Reviewer:
A very complete narrative review on filagrin, from its discovery ot its currrent implications in dermatology and atopic dermatitis; I really enjoyed reading it; only minor revisions.
Response: First of all, many thanks for this nice comment. We take it very gladly.
Minor comments:
In the text, the acronym for atopic dermatitis is not reported, just in the abstract...please check.
Response: the reviewer is totally correct. We have amended the manuscript accordingly.
Line 150, you should add: "Atopic dermatitis (AD), also known as atopic eczema, is one of the most common inflammatory skin diseases, affecting 15%-30% of children and up to 14.3% of adults, of whom approximately 20% have moderate-to-severe disease" and cite : doi: 10.18176/jiaci.0519. and doi: 10.1111/exd.14276.
Response: We have modified the manuscript as follows: AD, also known as atopic eczema, is the most common inflammatory skin diseases, affecting 1%-36% of children and up to 18% of adults, of whom approximately 20% have moderate-to-severe disease.
We have cited the two articles proposed by reviewer 2 and addition to 2 others.